# An ARMS-Multiplex PCR Targeting SARS-CoV-2 Omicron Sub-Variants

**DOI:** 10.3390/pathogens12081017

**Published:** 2023-08-06

**Authors:** Petros Bozidis, Eleni Petridi, Konstantina Gartzonika

**Affiliations:** Department of Microbiology, Faculty of Medicine, School of Health Sciences, University of Ioannina, 45110 Ioannina, Greece; bl02033@uoi.gr (E.P.); kgartzon@uoi.gr (K.G.)

**Keywords:** SARS-CoV-2, Omicron, variants, Multiplex PCR, ARMS, BA.1, BA.2, BA.4, BA.5

## Abstract

As of November 2021, the SARS-CoV-2 Omicron variant had made its appearance, gradually replacing the predominant Delta variant. Since its emergence, the Omicron variant has been continuously evolving through more than 500 strains, most of which belong to five sub-variants known as BA.1, BA.2, BA.3, BA.4, and BA.5. The aim of this study was to develop a multiplex polymerase chain reaction (PCR) that will be able to distinguish the basic sub-variants of Omicron in a rapid and specific way. Full genome sequences of Omicron strains with high frequency and wide geographical distribution were retrieved by the NCBI Virus and ENA databases. These sequences were compared to each other in order to locate single nucleotide polymorphisms common to all strains of the same sub-variant. These polymorphisms should also be capable of distinguishing Omicron sub-variants not only from each other but from previously circulating variants of SARS-CoV-2 as well. Thus, specific primers targeting characteristic polymorphisms of the four Omicron main branches BA.1, BA.2, BA.4, and BA.5 were designed according to the principles of the amplification refractory mutation system (ARMS) and with the ability to react under multiplex PCR conditions. According to our results, the ARMS-multiplex PCR could successfully distinguish all Omicron sub-variants that carry the corresponding mutations.

## 1. Introduction

Three years after the COVID-19 pandemic, the prevalent strains of SARS-CoV-2 worldwide belong almost exclusively to the Omicron variant. [1]. Its course begins with the emergence of the PANGO lineage B.1.1.529 in November 2021 and was designated by the World Health Organization’s (WHO) Technical Advisory Group on SARS-CoV-2 Virus Evolution with the Greek letter Omicron [2]. B.1.1.529 rapidly evolved into five more related branches, namely, BA.1 (B.1.1.529.1), BA.2 (B.1.1.529.2), BA.3 (B.1.1.529.3), BA.4, and BA.5 [3,4]. Although it is very difficult to find a consolidated record of Omicron sub-variants, it is estimated that more than 500 sub-variants and descendants have appeared so far [5].

Since the beginning of the pandemic, the WHO has established a surveillance and warning system that classifies circulating SARS-CoV-2 strains into two categories based on certain criteria associated with the degree of public health hazard. These categories include the Variants of Concern (VOC) and the Variants of Interest (VOI) [6]. On 7 June 2022, the WHO added a new category to its SARS-CoV-2 variant tracking system in order to deal with the increasing global transmission of the Omicron members and the anticipated rapid viral diversity among the Omicron descendants. This category was called VOC Lineages Under Monitoring (VOC-LUM) and has the role of informing public health authorities of VOC lineages that may require prioritized attention and monitoring [7]. Furthermore, because, already in February 2022, over 98% of all sequenced samples belonged to the Omicron family, the WHO updated its tracking system and working definitions for variants of SARS-CoV-2 on 15 March 2023 to better correspond to the current variant landscape that is shaped by the constant evolution and dominance of Omicron sub-lineages [8]. Therefore, it is still a daily challenge for laboratories around the world to identify the Omicron descendants in samples from patients with COVID-19.

To this purpose, we have developed a method for rapid detection and identification of the four key groups within the Omicron family. The method is based on the combination of multiplex polymerase chain reaction (PCR) and the amplification refractory mutation system (ARMS) [9], according to which specialized primers are designed in order to discriminate the Omicron sequences into four groups that include all sub-variants with a common origin.

## 2. Materials and Methods

### 2.1. Samples, RNA Isolation and Reverse Transcription

The samples used in this study came from COVID-19-positive subjects in Epirus, Greece, between January 2022 and April 2023, where linked whole genome sequencing (WGS) results confirmed infection with sub-variants BA.1, BA.2, BA.4, or BA.5. Total RNA was extracted from nasopharyngeal swabs using the KingFisherTM Flex Purification System (Life Technologies Holdings Pte. Ltd., Singapore) according to the manufacturer’s guidelines. Ten μL (0.3–1 μg) of the isolated RNA was reverse transcribed using the Invitrogen SuperScript II First–Strand Synthesis System (Thermo Fisher Scientific, Carlsbad, CA, USA), following the recommended protocol issued by the manufacturer. The production of viral genomic cDNA was tested with a commercial multiplex RT-qPCR assay for the detection of SARS-CoV-2, the GeneFinder COVID-19 Plus RealAmp Kit (Osang Healthcare Co., Ltd., Anyang-si, Republic of Korea), which targets the E, N, and RdRp genes. All RT-qPCR reactions were performed on a CFX96 Touch Real-Time PCR Detection System (Bio-rad Laboratories, Inc., Hercules, CA, USA).

### 2.2. Sequence Retrieval and Mutation Targeting

All SARS-CoV-2 nucleotide sequences were retrieved by the NCBI Virus database (https://www.ncbi.nlm.nih.gov/labs/virus/vssi/#/) (last time accessed on 29 May 2023) using the Pango lineage code name and setting the refinement results criteria for “sequence type” and “nucleotide completeness” to “GenBank” and “complete”, respectively. The fasta files were assigned to the Pangolin COVID-19 Lineage Assigner (https://pangolin.cog-uk.io) (last time accessed on 31 May 2023) in order to verify the lineage of the retrieved sequences. Sequences belonging to the same Omicron sub-lineage were compared using Clustal Omega (https://www.ebi.ac.uk/Tools/msa/clustalo/) (last time accessed on 16 June 2023). The same software was used for the comparison of sequences that belong to different sub-lineage groups.

### 2.3. Design of Primers and PCR

A combination of multiplex PCR with the ARMS method was applied for the discrimination of the four Omicron sub-variant groups. According to this method, specific primers targeting characteristic single nucleotide polymorphisms for each sub-variant were designed using Primer3web software (v.4.1.0) (https://bioinfo.ut.ee/primer3/) (accessed on 1 February 2023). To improve the specificity of ARMS primers, two mismatches were added at the third and fifth (or seventh) nucleotides from the 3′ end of the primer (Table 1). ARMS-Multiplex PCR reactions contained 5–10 μL of the reverse transcription reaction, 1× PCR buffer, 4.5 mM MgCl2, 200 μM dNTPs, 2.5 pmol/μL of each ARMS primer, 2.5 pmol/μL of each pair primer, and 2.5 U Taq DNA polymerase (Hot Start Taq DNA Polymerase, New England Biolabs, Ipswich, MA, USA) in a final volume of 25 μL. The following conditions were used to carry out the amplification reaction in a conventional thermocycler (PTC-200, Pelter Thermal Cycler, MJ Research, Inc., Waltham, MA, USA): 95 °C for 5 min, then 45 cycles comprised of 95 °C for 15 s, 58 °C for 15 s, and 68 °C for 45 s, followed by a final extension step at 68 °C for 5 min. At cycle 30, a pause was made in the thermocycling program, and 2.5 extra units of Taq were added for amplicon enhancement. The PCR products were analyzed by electrophoresis on a 1.5% (*w*/*v*) agarose gel stained with 10 μΜ ethidium bromide in 1 × TBE buffer. The electrophoresis results were observed under UV light. The size of each amplicon was estimated using a 100 bp DNA ladder (FastGene 100 bp DNA Marker, Nippon Genetics Europe, Düren, Germany), and the correct (or expected) molecular size of the amplified DNA products in electrophoresis was used for the detection of each sub-variant. Additionally, negative controls, such as RNAase and DNAase free water or variants other than Omicron, were used to rule out contamination or false positive reactions during the PCR step.

## 3. Results

### 3.1. Sequence Retrieval and Verification

There is a plethora of available public repositories for SARS-CoV-2 genomic data and variant tracking web resources (e.g., GISAID, NCBI, COG-UK, CoVariants, etc.) [10]. A search in these bases using the name of the strain assigned either by the WHO or the PANGO classification system as a keyword results in the appearance of sequences that significantly differ from each other, both in size and, in several cases, in the polymorphisms that they carry despite belonging to the same lineage. These differences may be due to differences in the quality of the samples, the various sequencing protocols and bioinformatics analysis methods used, or in the quality of the sequencing reactions themselves [10]. Thus, we used two reliable and popular databases in combination as a source for sequence retrieval, that of NCBI Virus and that of ENA (EMBL) using the full sequence size and readings that did not include ambiguous bases due to sequence errors as criteria. In order to ensure that the selected sequences for a specific primer design would be representative of the majority of the deposited sequences for each sub-variant, we proceeded with a two-stage verification process. The first step of this process involved the identity confirmation of the selected sequence by the pangolin tool, Pangolin COVID-19 Lineage Assigner, while the second step involved the comparison of the sequence using BLAST. Sequences that showed 100% success in the first step and at least 99% success in the second confirmation step were used for primer design, while the controversial ones were rejected (Figure 1). Through this process, we ended up using for subsequent comparison 19 sequences for BA.1 sub-variants, 34 sequences for BA.2 sub-variants, 12 sequences for BA.4 sub-variants and 32 sequences for BA.5 sub-variants for subsequent comparison (these sequences were retrieved until 31 January 2023).

### 3.2. Discrimination of Basic Omicron Sub-Variants by Multiplex ARMS-PCR

In order to locate common polymorphisms among all the sequences within each group, the verified sequences belonging to the same Omicron sub-variant were compared to each other and against the WUHAN sequence (NC_045512.2).These polymorphisms of each group were then compared to the ones of the other groups in order to find unique polymorphisms characteristic of each group. Eventually, this process resulted in 15 characteristic polymorphisms for the group of BA.1, one for the group of BA.2, one for the group of BA.4, and two for the group of BA.5 sequences (Appendix A). Since these polymorphisms span the viral genome, the comparison of the detected common polymorphisms for the Omicron groups was extended against other key SARS-CoV-2 variants such as alpha, beta, gamma, etc. This would ensure that the selected nucleotide positions could be used to distinguish not only the Omicron sub-variants from each other but also from the previous SARS-CoV-2 most predominant lineages.

The finally selected polymorphisms enabled us to design specific primers using the principles of the ARMS method [9]. These primers were tested in RNA samples originating from COVID-19 positive subjects in which the virus strain was previously identified by next-generation sequencing. The multiplex ARMS-PCR was tested against 15 BA.1, 18 BA.2, 1 BA.4, and 62 BA.5 sub-variants that were available in our laboratory and all gave specific reactions as shown in Figure 2. In addition, the ARMS primers were tested against the main previously circulating VOCs that were in our repository, namely B.1.1.7, B.1.351, B.1.617.2 and B.1.617.2/AY.1 and gave no cross reactions to any. All the ARMS primers of the multiplex PCR are depicted in Table 1.

### 3.3. The Potential of Omicron Specific Multiplex ARMS-PCR

Due to the extended genetic variation of the Omicron variant, we tried to explore the full potential of our method by comparing all the available sub-variant sequences against the polymorphisms we chose to design the ARMS primers. We were able to track the sequences of 342 such variants. These sequences were retrieved from the NCBI Virus and ENA databases and met the criteria of the two-stage verification process mentioned above. According to the nucleotide comparison data, the Omicron-specific multiplex ARMS-PCR has the potential to classify most of the sequences into sub-variant groups by the production of a specific amplicon (Appendix A). Exceptions to this rule will be observed in the few cases where recombinant variants originated by BA.1 and BA.2 (e.g., XG, XL, XN) or BA.2 and BA.5 (e.g., XBD, XBG, XBJ) descendants’ recombination is tested. Therefore, a guide for the potent implementation of the method is given regarding the majority of sub-variants that have appeared in the past and continue to circulate worldwide.

## 4. Discussion

Since their emergence, all Omicron sub-variants have exhibited increased contagiousness and a remarkable ability to escape the host’s immune system compared to their predecessors, especially the Delta sub-variants, which they replaced until the end of 2021 [11]. The Omicron variant gave rise to a large number of new mutants based on both the high nucleotide mutation rate of SARS-CoV-2 in general [12] and the genetic recombination events that took place within a population made mainly of immune individuals [13]. Although the main five sub-lineages, namely BA.1, BA.2, BA.3, BA.4, and BA.5, started to emerge consecutively in early 2022, they continued to circulate simultaneously worldwide, even through their descendants, until recently, in a phenomenon that has been called “variant soup” [14]. Within this environment, the generation of new sub-variants through genetic recombination between strains originating from a single branch has been decisive for the evolution of the Omicron group. Thus, although the Omicron strains designated “XD” and “XF” were derived from BA.1 and Delta variants, other strains like “XE” were recombination products between BA.1 and BA.2 Omicron sub-lineages [15,16].

As of March 2023, the European Centre for Disease Prevention and Control (ECDC) has de-escalated BA.2, BA.4, and BA.5 from its list of SARS-CoV-2 variants of concern (VOC) as these parental lineages are no longer circulating [17]. However, some of the new SARS-CoV-2 variants are derived from the recombination between at least two distinct parental Omicron lineages or sub-lineages within the same Omicron lineage [18]. Thus, a number of co-circulating BA.2 and BA.5 descendent variants are listed by ECDC either as variants of interest (VOI), among which BQ.1 (BA.5 descendent), BA.2.75, XBB (BA.2.10.1 /BA.2.75 descendent), and XBB.1.5 (BA.2.10.1/BA.2.75 descendent), or as variants under monitoring (VUM), among which BF.7 (BA.5 descendent), BA.2.3.20 (BA.2 descendent), CH.1.1 (BA.2.75 descendent), BN.1 (BA.2.75 descendent), XBC [Delta (21I)/BA.2 recombinant], and XAY [Delta (AY.45)/BA.2 recombinant] [17]. The most recent SARS-CoV-2 recombinant is the Omicron XBB lineage with its descendant XBB.1.5 sublineage, which is generated by the recombination of two distinct BA.2 sublineages [1,17,18]. According to a global epidemiological report, XBB.1.5 prevalence has been declining steadily, while XBB.1.16, a variant very closely related to XBB.1.5, has shown increasing trends [1]. Neither XBB.1.5 nor XBB.1.16 possess the characteristics to become a global threat, although XBB.1.16 has a greater growth advantage in the human population compared with XBB.1.5 [19,20]. Regarding the main BA Omicron groups, there are a few reports that try to explore the differences between the BA.2 and BA.5 lineages in terms of reinfection rate and disease severity. In a nationwide population-based study in Denmark, Hansen and colleagues found greater odds of hospitalization for patients infected with BA.5 compared to BA.2 [21]. Their results were similar to the ones generated by Kang and colleagues in England, according to which individuals infected with the BA.5 sub-lineage exhibited more severe symptoms at the onset of symptomatic disease [22]. On the contrary, another cohort study by Aziz et al. reported no differences in the risk of hospital admission or death or in oxygen supplementation following presentation to an emergency department for BA.4 or BA.5 patients compared to BA.2 patients [23]. In agreement with the latter conclusion were also the results presented by other research groups for BA.4 and BA.5 [24,25] or XBB variants [26].

Although it is not quite clear whether the Omicron sub-variants differ from each other in the severity of the clinical outcome, local diagnostic laboratories still need to constantly monitor their epidemiology. Due to their continuous evolution and extensive simultaneous spread worldwide, there are very limited molecular detection tests available that are also capable of distinguishing specific Omicron sub-variants [27,28,29,30]. In some cases, the escape from molecular detection, by S dropout (the so-called “S gene target failure” or SGTF) has been related to the detection of BA.4 and BA.5 sub-lineages. However, SGTF may be used only as a pre-screening procedure with low discrimination capacity, as the majority of Omicron and some non-Omicron variants (e.g., Alpha) also give an SGTF result. [31]. In a recent study by Chrysostomou et al., the identification of Omicron sub-variants was achieved through the detection of a combination of deletions/insertion in the ORF1a and S gene regions using a uniplex molecular beacon-based real-time RT-PCR assay. This assay correctly identified the presence of specific targets; however, some Omicron variants, such as BA.4 and BA.5, couldn’t be distinguished from each other because they displayed the same pattern of results [32]. The same limitation in distinguishing Omicron variant groups is also seen in another recent study that is based on single-nucleotide polymorphism analysis, where only BA.1 and BA.2 sub-variants were included in the study [33].

In our study, multiplex PCR in combination with the ARMS method could be used to distinguish different Omicron sub-variants from each other and from previously appearing variants in a fast, reliable, and economical way. We did not include the BA.3 sub-variant in our analysis since its transmissibility was very limited, with only a few hundred cases at most globally. Additionally, BA.3 falls under the overall Omicron clade 21M and does not stand as a separate branch according to Nextstrain clades since it does not yet satisfy the Nextstrain clade definition criteria [3,34]. This method has been tested, and has proven to be effective, on several Omicron strains available in our lab as well as on previous variants that have already disappeared or are still circulating in the community.

However, some details that we consider critical for the better application of the method in terms of its sensitivity and specificity should be noted. As far as the sensitivity of the method is concerned, two elements should be taken into consideration: a) cDNA obtained from a patient’s sample that has been tested positive for SARS-CoV-2 by RT-qPCR with a threshold cycle (Ct) less than 30 should be used, and b) at least 5–10 μL of the reverse transcription reaction (~500 ng cDNA/reaction minimum) should be used. On the other hand, the specificity of the method is ensured by two factors: a) a Taq polymerase without proofreading capability must always be used; b) the amplification reaction should be performed in the presence of a high concentration of Mg^2+^ (4–4.5 mM) and low concentration of primer mix (1–2.5 pmol/μL).

We have used the principles of ARMS in the past for allelic discrimination and they have proven to be effective and adaptive [35]. In this case, based on the same principles, we developed a reliable method that displays high specificity in distinguishing the Omicron sub-variants. Primers were designed based on single nucleotide polymorphisms that were characteristic of each Omicron variant group to ensure that they would be representative of the majority of the deposited sequences. Furthermore, a list based on the deposited sequences is provided, according to which the method could be applied against 342 Omicron sub-variants and 14 non-Omicron variants. Thus, although the method is not able to recognize the initial Omicron strain, B.1.1.529 (which has already disappeared), it could be used for the detection of the circulating SARS-CoV-2 variants as well. As the evolutionary landscape of the virus is dynamic and rapidly changing and new strains are constantly emerging, we believe the approach we present could be useful for the development of molecular diagnostic and genotyping techniques through the selection of appropriate primers and probes. So, whenever new emerging variants or sub-lineages are detected, the test could potentially be extended with the development of new specific primers. Our method is an in-house diagnostic tool that is simple but efficient and can be implemented in a basically equipped laboratory as a pre-screening method to discriminate distinct variant groups especially when either commercial solutions or sophisticated genomic monitoring are not available.

## Figures and Tables

**Figure 1 pathogens-12-01017-f001:**
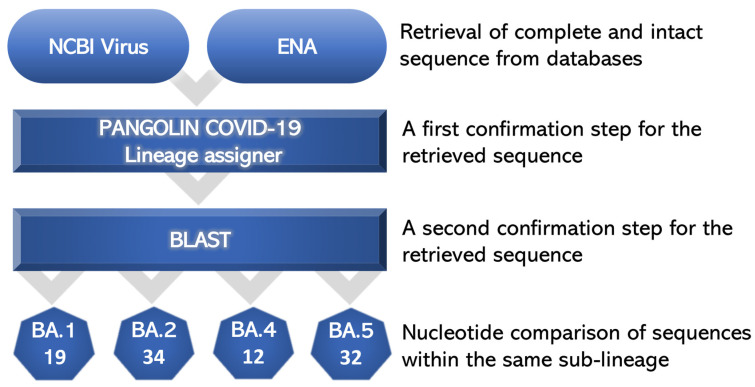
Sequences were retrieved from the NCBI Virus and ENA databases. Selected sequences were confirmed by the Pangolin COVID-19 Lineage assigner and BLAST tools. Only sequences that were verified by 100% and 99%, respectively, were selected for nucleotide comparison within the same sub-variant group.

**Figure 2 pathogens-12-01017-f002:**
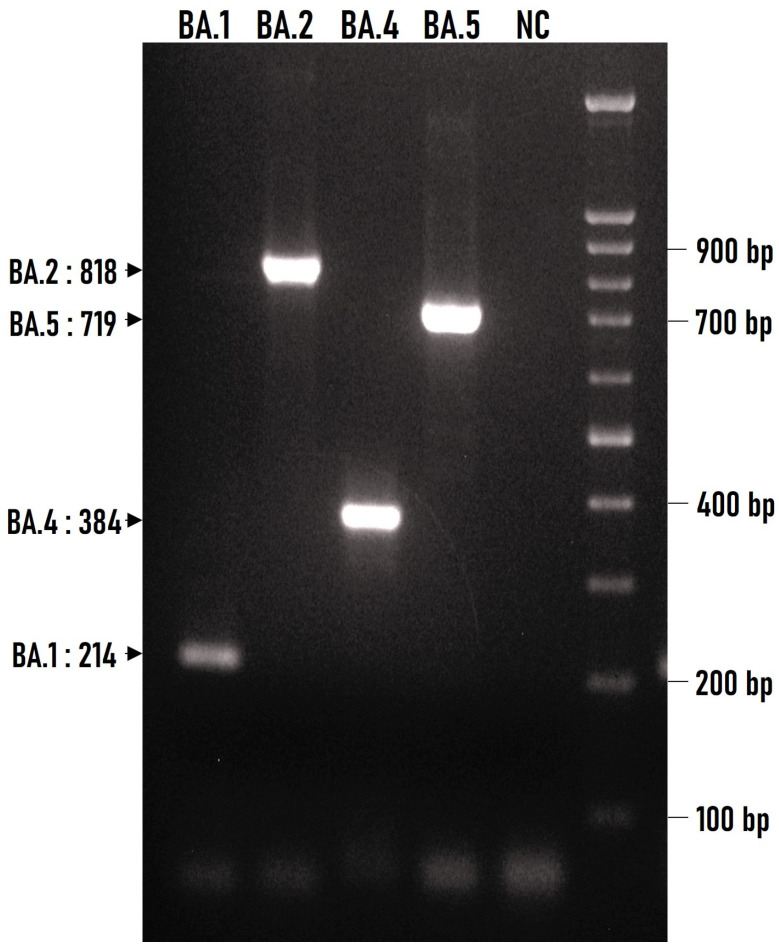
Agarose gel electrophoresis (1.5% *w*/*v*) of ARMS–multiplex PCR products generated by the amplification of four cDNAs corresponding to the four Omicron (BA.1, BA.2, BA.4, and BA.5) sub-variants. A 100 bp DNA ladder was used to confirm the correct length of the anticipated amplicons. The size of each amplicon is indicated on the left. NC, negative control (without DNA).

**Table 1 pathogens-12-01017-t001:** Primers used in ARMS multiplex PCR. The mutated nucleotides targeted by the ARMS primers are noted with red color while additional mismatches introduced at the 3′end of the ARMS primer are underlined and noted with green color.

Sub-Variant	Mutation and Genetic Locus	Direction	Primer	Position on Wuhan (NC_045512.2)	Sequence	PCR Product (bp)
BA.1	A to G (2832) Orf1ab	Forward	2832 ARMS	2809–2832	GAT TGA TAA AGT ACT TCA TGC GAG	214
		Reverse		3022–2999	ATG TGA AGC CAA TTT AAA CTC ACC	
BA.2	C to T (9866) Orf1ab	Forward	9866 ARMS	9842–9866	TTG CGT AGT GAT GTG CTC TTA CAT T	818
		Reverse		10,659–10,637	ACT GTA ATA GTT GTG TCC GTA CC	
BA.4	G to T (27,788) Orf7a-Orf8 interspace	Forward	27,788 ARMS	27,762–27,788	GAA CTT TCA TTA ATT GAC TAC TAT CTT	384
		Reverse		28,145–28,119	TAA ACA GGA AAC TGT ATA ATT ACC GAT	
BA.5	C to T (26,529) M gene	Forward		25,835–25,859	GCT GGC ATA CTA ATT GTT ACG ACT A	719
		Reverse	26,529 ARMS	26,553–26,529	CAA CGG TAA TAG TAC CGT GGG ATT T	

## Data Availability

Not applicable.

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
