# Peer review of "An ARMS-Multiplex PCR Targeting SARS-CoV-2 Omicron Sub-Variants"

_pathogens, 2023, doi:10.3390/pathogens12081017_

Round 1
Reviewer 1 Report
The authors discuss the emergence and evolution of the Omicron variant of SARS-CoV-2, which has given rise to more than 500 strains, including sub-variants. The study's objective was to develop a multiplex polymerase chain reaction (PCR) assay capable of rapidly and specifically distinguishing basic sub-variants of SARS-CoV-2 (BA.1, BA.2, BA.3, BA.4, and BA.5). Full genome sequences of high-frequency Omicron strains from diverse locations were retrieved from the NCBI Virus and ENA databases for analysis. By identifying common single nucleotide polymorphisms (SNPs) shared among strains of the same sub-variant, the researchers aimed to differentiate Omicron sub-variants not only from each other but also from previously circulating SARS-CoV-2 variants. Specific primers were designed to target characteristic polymorphisms of the main branches (BA.1, BA.2, BA.4, and BA.5) using the amplification refractory mutation system (ARMS) principles for multiplex PCR. The results indicate that the ARMS-multiplex PCR successfully distinguished all Omicron sub-variants carrying the corresponding mutations.
Minor suggestions:
-Abstract: “As of November 2021…….” Sounds like misleading information about the variants circulating by then.
-Line: 109- repetition of the wordings.
-Results of the analysis did not represent any statistical information.
-Complete sequences of the variants (patient isolates) may be useful for confirmation.
- Please comment on the novel aspects of the study.
Manuscript contains a few typographical errors.
Author Response
REVIEWER 1
Quality of English Language
( ) I am not qualified to assess the quality of English in this paper
( ) English very difficult to understand/incomprehensible
( ) Extensive editing of English language required
(x) Moderate editing of English language required
( ) Minor editing of English language required
( ) English language fine. No issues detected
|
Yes |
Can be improved |
Must be improved |
Not applicable |
|
|
Does the introduction provide sufficient background and include all relevant references? |
( ) |
(x) |
( ) |
( ) |
|
Are all the cited references relevant to the research? |
( ) |
(x) |
( ) |
( ) |
|
Is the research design appropriate? |
(x) |
( ) |
( ) |
( ) |
|
Are the methods adequately described? |
( ) |
(x) |
( ) |
( ) |
|
Are the results clearly presented? |
( ) |
(x) |
( ) |
( ) |
|
Are the conclusions supported by the results? |
(x) |
( ) |
( ) |
( ) |
Comments and Suggestions for Authors
The authors discuss the emergence and evolution of the Omicron variant of SARS-CoV-2, which has given rise to more than 500 strains, including sub-variants. The study's objective was to develop a multiplex polymerase chain reaction (PCR) assay capable of rapidly and specifically distinguishing basic sub-variants of SARS-CoV-2 (BA.1, BA.2, BA.3, BA.4, and BA.5). Full genome sequences of high-frequency Omicron strains from diverse locations were retrieved from the NCBI Virus and ENA databases for analysis. By identifying common single nucleotide polymorphisms (SNPs) shared among strains of the same sub-variant, the researchers aimed to differentiate Omicron sub-variants not only from each other but also from previously circulating SARS-CoV-2 variants. Specific primers were designed to target characteristic polymorphisms of the main branches (BA.1, BA.2, BA.4, and BA.5) using the amplification refractory mutation system (ARMS) principles for multiplex PCR. The results indicate that the ARMS-multiplex PCR successfully distinguished all Omicron sub-variants carrying the corresponding mutations.
Minor suggestions:
-Abstract: “As of November 2021…….” Sounds like misleading information about the variants circulating by then.
AUTHORS RESPONSE:
The initial sentence, “As of November 2021, the predominant variant of SARS-CoV-2 worldwide is Omicron”, was rephrased as follows:
“As of November 2021, the SARS-CoV-2 Omicron variant had made its appearance, gradually replacing the predominant Delta variant.”
We would like to thank the reviewer for his comment.
-Line: 109- repetition of the wordings.
AUTHORS RESPONSE:
The double verb “differ” in the sentence was erased.
We would like to thank the reviewer for his comment.
-Results of the analysis did not represent any statistical information.
AUTHORS RESPONSE:
It is not so clear to us in which part of our work statistical data processing should be engaged. If the reviewer means the number of samples we used to test the specific method and the success rate we had for each individual category, we could include these data in the text according to the following table:
BA.1: 15
BA.2: 18 (ΒΑ.2: 8, ΒΑ.2.9: 4, ΒΑ.2.3: 1, ΒΑ.2.12.1: 1, ΒΑ.2.75+: 1, BR.2.1: 1, ΧΒΒ.1.5: 1, ΒΝ.1.5: 1)
BA.4: 1
BA.5: 62 (ΒΑ.5.9: 7, ΒF.5: 3, BF.7: 9, BF.7.6: 1, BA.5.2.1: 7, BA.5.2.21: 3, ΒΑ.5.2: 5, ΒΑ.5.1: 4, ΒΑ.5.2.34: 1, ΒΑ.5.2.20: 1, ΒΑ.5.2.12: 1, ΒΑ.5.1.5: 1, BQ.1: 1, BQ.1+: 2, BF.14: 1, CK.2.1.1: 1, BQ.1.8: 2, BQ.1.1.13: 1, BQ.1.1.8: 1, BQ.1.1: 7, ΒΑ.5.1.23: 1, ΒΑ.5.2.7: 1, ΒΑ.5.1.10: 1
Since the method worked successfully for all the identified by next-generation sequencing samples that we had available in our lab, we considered that the numerical data from these tests would not provide any additional information. This is why we make no more than a simple statement between lines 156-160 in the text that all our tests were positive. However, we have rephrased the sentence: “We tested the ARMS primers in a large number of Omicron sub-variants that were available in our laboratory, and all gave specific reactions as shown in Fig.2.” (lines 158–160) as follows:
“The multiplex ARMS-PCR was tested against 15 BA.1, 18 BA.2, 1 BA.4, and 62 BA.5 sub-variants that were available in our laboratory and all gave specific reactions as shown in Fig.2”.
In any case, we think that the nature of our experiments did not require any statistical analysis.
We would like to thank the reviewer for his comment.
-Complete sequences of the variants (patient isolates) may be useful for confirmation.
AUTHORS RESPONSE:
As we mention in the text (lines 59–62), the information regarding the identification by NGS of the SARS-CoV-2 sub-variants was linked to the samples we used in this study, meaning that we did not perform the sequencing by our own means. Since the beginning of the pandemic, our Laboratory has been participating in the Greek National SARS-CoV-2 Genomic Surveillance Network as a SARS-CoV-2 Reference Center for the region of Epirus and the Ionian Islands, Western Greece. To date, we have analyzed more than 240,000 clinical samples collected from suspected SARS-CoV-2 patients in that region. During this course, we used to send a number of samples of clinical importance to the Hellenic National Health Organization – EODY in a weekly basis. NGS analysis was conducted in their facilities, and a report was sent back to the Reference Center as a standard procedure. As you realize, we only have the NGS identification of the variants that we have used in this work and not the complete sequences per se. However, we have no doubt that the sequencing information provided to us by the Hellenic National Health Organization was, in any case, correct. After all, our method with primers designed based on sequences retrieved by public databases responded well in all cases examined, which is consistent with the sequencing data we had in hand. We would also like to add that, due to our active participation in the National SARS-CoV-2 Genomic Surveillance Network, we possess in our collection human samples infected with almost all the SARS-CoV-2 variants that have been circulating in Greece and have been identified by NGS. Based on our expertise in SARS-CoV-2 diagnosis and our broad collection of identified SARS-CoV-2 variants, we were able to publish our observations concerning the so called "drop out" effect during the molecular detection of specific variants (B.1.1.318) using specific commercial RT-PCR kits (1).
- Gartzonika, K.; Bozidis, P.; Priavali, E.; Sakkas, H. Rapid Detection of blaKPC-9 Allele from Clinical Isolates. Pathogens 2021, 10, 487. doi: 10.3390/pathogens10040487.
We would like to thank the reviewer for his comment.
- Please comment on the novel aspects of the study.
AUTHORS RESPONSE:
The following comment was added in the last paragraph of the manuscript.
“As the evolutionary landscape of the virus is dynamic and rapidly changing and new strains are constantly emerging, we believe the approach we present could be useful for the development of molecular diagnostic and genotyping techniques through the selection of appropriate primers and probes. So,.... Our method is an in-house diagnostic tool that is simple but efficient and can be implemented in a basically equipped laboratory as a pre-screening method to discriminate distinct variant groups especially when either commercial solutions or shophisticated genomic monitoring are not available.”
We would like to thank the reviewer for his comment.
Comments on the Quality of English Language
Manuscript contains a few typographical errors.
AUTHORS RESPONSE:
The manuscript has been thoroughly checked and typographical errors have been corrected.
We would like to thank the reviewer for his comment.

Reviewer 2 Report
Review of Bozidis et al
In this study, the authors developed a multiplex PCR method to distinguish the sub-variants of Omicron. In their results, the authors tested the ARMS primers and showed they could specifically PCR Omicron BA.1, BA.2, BA.4 and BA.5. This is an interesting work, which
could help to distinguish Omicron sub-strains. However, in view of the limited data and findings of this study, the flaws in the experimental design and timeliness of this work, the reviewer's passion to this study is greatly reduced.
Major issues
1. In figure 2, the authors showed that different primers could PCR very specific bands for each Omicron sub-variants. But they did not show whether these primers could cross reactive to other variants? For instance, can BA.1 primers work for BA.4/BA.5? They should test all other variants with each of the primer and confirm their specificity.
2. Now Omicron XBB variants (descendent variants) are dominating the pandemic, while BA.1, BA.2, BA.3, BA.4 and BA.5 variants are almost disappeared. Can the authors also design primers specific for XBB variants (e.g., XBB1.5, XBB1.16) and see whether these primers can also distinguish from other Omicron sub-variants?
3. In this paper, the authors cited lots of references which focused on Omicron BA series variants (mostly 2022 and early 2023), which is not the case now. As a result, this study cannot provide useful help for the current epidemic situation. The authors should edit their whole text with updated references and add new data as mentioned in Major issue NO. 2.
Minor comments
1. The writing in general was not great, need more editing. There were also some typos/grammar mistakes in the text, they should go over and fix them.
2. In figure 1, please use same font size and try to avoid multiple colors.
3. In table 1, please change the font to make it easier for the reviewer/readers to read.
4. In figure 2, please remove “PCR product length (bp)” and “100 bp DNA ladder”.
5. Line 193-194, is this true for now? Please update reference.
6. Identification of Omicron variants can be easily done by deep sequencing, why the authors chose PCR? They should introduce/discuss the advantages of their method compared to NGS.
See comments.
Author Response
REVIEWER 2
Quality of English Language
( ) I am not qualified to assess the quality of English in this paper
( ) English very difficult to understand/incomprehensible
( ) Extensive editing of English language required
(x) Moderate editing of English language required
( ) Minor editing of English language required
( ) English language fine. No issues detected
|
Yes |
Can be improved |
Must be improved |
Not applicable |
|
|
Does the introduction provide sufficient background and include all relevant references? |
( ) |
(x) |
( ) |
( ) |
|
Are all the cited references relevant to the research? |
( ) |
(x) |
( ) |
( ) |
|
Is the research design appropriate? |
( ) |
( ) |
(x) |
( ) |
|
Are the methods adequately described? |
( ) |
( ) |
(x) |
( ) |
|
Are the results clearly presented? |
( ) |
( ) |
(x) |
( ) |
|
Are the conclusions supported by the results? |
( ) |
( ) |
(x) |
( ) |
Comments and Suggestions for Authors
Review of Bozidis et al
In this study, the authors developed a multiplex PCR method to distinguish the sub-variants of Omicron. In their results, the authors tested the ARMS primers and showed they could specifically PCR Omicron BA.1, BA.2, BA.4 and BA.5. This is an interesting work, which could help to distinguish Omicron sub-strains. However, in view of the limited data and findings of this study, the flaws in the experimental design and timeliness of this work, the reviewer's passion to this study is greatly reduced.
Major issues
1. In figure 2, the authors showed that different primers could PCR very specific bands for each Omicron sub-variants. But they did not show whether these primers could cross reactive to other variants? For instance, can BA.1 primers work for BA.4/BA.5? They should test all other variants with each of the primer and confirm their specificity.
AUTHORS RESPONSE:
With all due respect to the reviewer's opinion, we believe that his comment is the result of a misunderstanding. The results shown in Figure 2 represent separate multiplex PCR reactions in which different cDNAs from each of the four variants is used as a template each time. As indicated in the text, these cDNAs are derived from human samples of patients infected with the virus where the variant has been identified by NGS. In each of these reactions, apart from the different cDNAs, all other reagents are the same, which means that in each reaction we add all eight primers, the four non-specific and the four specific (ARMS) ones, which we have described in Figure 1. Hence, in each reaction depicted in Figure 2, we test the generation of the specific product for the corresponding strain (e.g., BA.1), as well as the response of the remaining six primers, in the presence of a template for which they are not expected to give any specific or non-specific reaction. Therefore, according to Figure 2, the specific primers for each strain produce the expected product, while they do not produce specific or non-specific reactions when they are present in reactions with templates for which they are not designed, all under the PCR reaction conditions that are described in detail in the text.
We would like to thank the reviewer for his comment.
Now Omicron XBB variants (descendent variants) are dominating the pandemic, while BA.1, BA.2, BA.3, BA.4 and BA.5 variants are almost disappeared. Can the authors also design primers specific for XBB variants (e.g., XBB1.5, XBB1.16) and see whether these primers can also distinguish from other Omicron sub-variants?
AUTHORS RESPONSE:
The reviewer’s comment is quite valuable. The evolutionary landscape of SARS-CoV-2 is changing rapidly. The aim of this study was not to distinguish all the Omicron variants from each other. It is suggested that there are over 500 Omicron variants, but this number is not easily identifiable. We have been able to identify 342 of them. The initial aim of this study was to be able to identify the clades of Omicron to provide information to clinicians in case the clades showed different clinical characteristics. We believe that our study successfully serves the original purpose we set out to achieve. The XBB variants referred to by the reviewer belong to clade BA.2 and can be identified by the method at the level for which this method was designed. The XBB variants appeared at the end of 2022 (1) and circulated in parallel with variants of the other Omicron clades. No one could predict which variant would eventually dominate. After all, the purpose of this identification method was to group the variants by finding polymorphisms that unite these variants into their groups, not to distinguish them from each other based on the different polymorphisms they possess. Thus, as is evident from the supplementary material that follows the text, there are only a few such common mutations (one or two at most) for all clades, as opposed to BA.1. In the text we describe the strategy that we have followed in detail and consider that the table that depicts these common mutations is an excellent source of information for future studies on Omicron clades. Thus, designing individual primers that differentiate the variants of XBB is a goal beyond and outside of the original study design.
We would also like to add that in our opinion this study is in no way obsolete according to the latest epidemiological data as a method of initial identification. Here are the most recent epidemiological data published on July, 2023 by WHO (2) and summarized in the table below:
Please note above that although the majority of predominant variants are descendants of the BA.2 lineage (all XBB subvariants, BA.2.75, CH.1.1), there is still an important increasing percentage (9.5%) of the so-called “unassigned” lineages in which all the rest of the Omicron sub-variants are included. This percentage is increasing during the latest weeks of 2023, and no one can predict what the landscape will be like in two months from now. Therefore, we believe that the value of this work lies in the fact that it can be a guide for future development of other tests targeting the discrimination of those variants that will end up dominating in the future.
- Tamura, T., Ito, J., Uriu, K. et al. Virological characteristics of the SARS-CoV-2 XBB variant derived from recombination of two Omicron subvariants. Nat Commun 14, 2800 (2023). https://doi.org/10.1038/s41467-023-38435-3
- Weekly epidemiological update on COVID-19 - 20 July 2023 Edition 152. https://www.who.int/publications/m/item/weekly-epidemiological-update-on-covid-19---20-july-2023.
We would like to thank the reviewer for his comment.
In this paper, the authors cited lots of references which focused on Omicron BA series variants (mostly 2022 and early 2023), which is not the case now. As a result, this study cannot provide useful help for the current epidemic situation. The authors should edit their whole text with updated references and add new data as mentioned in Major issue NO. 2.
AUTHORS RESPONSE:
This study is not intended to record the current epidemic situation. It is not our intention to describe the evolution of the virus currently. This work describes a practical and easy way to genotype the four clades of Omicron as they have been defined by phylogenetic relationships. It could also be a retrospective study that would provide a guide that could be used in order to distinguish the predominant SARS-CoV-2 variants. Nevertheless, as stated in our previous answer, the work is not only about the past but also about the discrimination of variants that still exist. Therefore, the literature we present is representative of the Omicron clades and the techniques used to distinguish them, and according to our knowledge, there are no other sources in the bibliography that deal with the subject of this study. Furthermore, we believe that we have listed interesting information about the latest circulating variants that can be initially classified in one of the Omicron clades by the method we propose in the supplementary material. Beyond this, in the discussion, we have already tried to give a brief overview of the current epidemiological situation by providing information on the circulating strains. However, considering the recommendations of the reviewer, we have replaced reference [1] with a more recent one, deleted reference [22], and finally added six more references, which update our references and contain more information on the current situation.
We would like to thank the reviewer for his comment.
Minor comments
The writing in general was not great, need more editing. There were also some typos/grammar mistakes in the text, they should go over and fix them.
AUTHORS RESPONSE:
We have gone through the manuscript and corrected grammar mistakes. All the changes have been underlined.
We would like to thank the reviewer for his comment.
In figure 1, please use same font size and try to avoid multiple colors.
AUTHORS RESPONSE:
We have made the font uniform both in the schemes and in the text and used the same font size (Abadi 17). Also, we have replaced the multiple colors with only one.
We would like to thank the reviewer for his comment.
In table 1, please change the font to make it easier for the reviewer/readers to read.
AUTHORS RESPONSE:
The font has changed and the colors inside of most of the cells have been erased. We believe these changes make the content more friendly to the reader.
We would like to thank the reviewer for his comment.
In figure 2, please remove “PCR product length (bp)” and “100 bp DNA ladder”.
AUTHORS RESPONSE:
All the above indications have been removed from figure 2.
We would like to thank the reviewer for his comment.
Line 193-194, is this true for now? Please update reference.
AUTHORS RESPONSE:
The landscape of the virus’s evolution is indeed changing with a glance of the eye. The reviewer may be right that during the last month the BA.2 descendants have been most of, or almost the only of the Omicron circulating variants, although one month ago the WHO reports showed that this was not the case since members from the BA.5 clade were listed as well. In addition, as proved by the data displayed in our response to the reviewer’s second major comment, there are still almost 10% of the circulating variants that belong possibly to other Omicron clades besides BA.2 and are registered as “unassigned”. As far as the short article by Ewen Callaway is concerned, we think that it is a very important report. Although it was published in Nature on November 10, 2022, it describes in a very elegant and brilliant way what many virologists have called the Omicron era, which we can briefly describe as the second phase of the COVID-19 pandemic, quite distinct both in terms of clinical and epidemiological characteristics as well as in evolutionary characteristics of the virus. This is why we like this publication so much and think it is necessary in our discussion, even though it talks about the evolutionary landscape of the Omicron variant only a year after its appearance.
However, in order to reflect the current epidemiological situation more accurately, we rephrased the disputed sentence as follows:
[Although the main five sub-lineages, namely BA.1, BA.2, BA.3, BA.4, and BA.5, started to emerge consecutively in early 2022, they continued to circulate simultaneously worldwide, even through their descendants, until recently, in a phenomenon that has been called “variant soup”.]
We would like to thank the reviewer for his comment.
Identification of Omicron variants can be easily done by deep sequencing, why the authors chose PCR? They should introduce/discuss the advantages of their method compared to NGS.
AUTHORS RESPONSE:
There is no doubt that deep sequencing is the ultimate tool for the identification of viral strains. However, a lot of clinical laboratories around the world do not have this capability as a routine procedure for all the clinical samples they examine. At this point, we would like to explain the reasons that led us to develop this method. The main reason was that we are a clinical university microbiology laboratory whose purpose is to provide early diagnosis and to support clinicians in their therapeutic work. During the pandemic, we always received questions from clinicians regarding the type of viral strain, especially in severe clinical cases. The same happened when the Omicron variants started to appear. As far as the earlier strains were concerned, there were usually commercial kits available with which we could at least indirectly identify the patient's viral strain. However, after the appearance of Omicron, we had no way of identifying the sub-variant. It should be noted that we do not have the ability to do direct next generation sequencing for all samples of clinical interest, which, if it were to happen, would be astonishing. We therefore proceeded to develop this method by applying the principles of ARMS, which we with success had used for allelic discrimination in the past (1). Οther research groups have used similar methods as a first approach for rapid identification of viral strains, including SARS-CoV-2 and even using the ARMS principles that we have used (2-4). The reasons for introducing such methods that offer less analytical capability are akin ours. According to the reviewer’s suggestion, we have tried to describe some of these reasons by adding text in the last paragraph of the discussion of the manuscript.
- Gartzonika K, Bozidis P, Priavali E, Sakkas H. Rapid Detection of blaKPC-9 Allele from Clinical Isolates. Pathogens. 2021;10(4):487. Published 2021 Apr 17. doi:10.3390/pathogens10040487.
- Gao L, Zu X, Liu X, Yu Z, Du Z, Hu Z, Xue Y. Establishment of a Rapid Typing Method for Coronavirus Disease 2019 Mutant Strains Based on PARMS Technology. Micromachines (Basel). 2022 Jan 18;13(2):145. doi: 10.3390/mi13020145. PMID: 35208270; PMCID: PMC8879704.
- Xiong D, Zhang X, Shi M, Wang N, He P, Dong Z, Zhong J, Luo J, Wang Y, Yu J, Wei H. Developing an Amplification Refractory Mutation System-Quantitative Reverse Transcription-PCR Assay for Rapid and Sensitive Screening of SARS-CoV-2 Variants of Concern. Microbiol Spectr. 2022 Feb 23;10(1):e0143821. doi: 10.1128/spectrum.01438-21. Epub 2022 Jan 5. PMID: 34985323; PMCID: PMC8729772.
- Islam MT, Alam ARU, Sakib N, Hasan MS, Chakrovarty T, Tawyabur M, Islam OK, Al-Emran HM, Jahid MIK, Anwar Hossain M. A rapid and cost-effective multiplex ARMS-PCR method for the simultaneous genotyping of the circulating SARS-CoV-2 phylogenetic clades. J Med Virol. 2021 May;93(5):2962-2970. doi: 10.1002/jmv.26818. Epub 2021 Feb 1. PMID: 33491822; PMCID: PMC8014803.
We would like to thank the reviewer for his comment.

Round 2
Reviewer 1 Report
The manuscript may be accepted for publication in its current form.
Reviewer 2 Report
The authors have addressed my concerns and this paper is suggested for publication.